# Interlayer Coupling and High-Frequency Performance in Magnetic Anisotropic FeCoB/Hf/FeCoB Trilayers with Various Hf Thicknesses

**Duo Liu** [1,2,†], **Shouheng Zhang** [1,†] **and Shandong Li** [1,2,*]

1    Center for Marine Observation and Communication, College of Physics, Qingdao University, Qingdao 266071, China; 18203996685@163.com (D.L.); zhangshouheng@126.com (S.Z.)
2    College of Electronics and Information, Qingdao University, Qingdao 266071, China
*    Correspondence: lishd@qdu.edu.cn
†    These authors contributed equally to this work.

**Abstract:** FeCoB (25 nm)/Hf($t_{Hf}$)/FeCoB (25 nm) sandwich films with different hafnium thicknesses $t_{Hf}$ were fabricated using a modified compositional gradient sputtering method to obtain self-biased high-frequency performances. The effects of $t_{Hf}$ on the interlayer coupling and FMR frequency were investigated. It is revealed that interlayer coupling enhanced the resonance frequency by 48%, and a ferromagnetic coupling between the FeCoB films occurred for the trilayers with $t_{Hf}$ < 3.0 nm, likely due to the interface roughness and pinhole effect. In this case, only acoustic mode resonance was observed with the same high-frequency performance as the corresponding FeCoB single layer. In contrast, a $t_{Hf}$-dependent antiferromagnetic interlayer coupling appeared at $t_{Hf}$ > 3.0 nm. The coupling coefficient $J_1$ was antiferromagnetic, and a biquadratic coupling $J_2$ appeared at $t_{Hf}$ > 3.5 nm. The coupling mechanism was simulated and verified by Layadi's rigid model, and the simulation was consistent with the experimental results.

**Keywords:** ferromagnetic resonance; magnetic anisotropy; interlayer exchange coupling; acoustic resonance; optical mode resonance

## 1. Introduction

A soft magnetic film (SMF) is one of the most useful materials in integrated circuits devices [1–6]. For the SMF, permeability $\mu$ and ferromagnetic resonance (FMR) frequency $f_r$ are the most important parameters since the electromagnetic components, such as high-frequency sensors, micro-inductors, magnetic recording heads, microwave noise filters, micro-transformers, and many other devices, need increasingly higher operation speed [7–11]. Kittel's equation adequately describes the $f_r$ in a self-biased field [4].

$$f_r = \frac{\gamma}{2\pi}\sqrt{H_K \cdot (H_K + 4\pi M_S)},\tag{1}$$

where $H_K$, $4\pi M_S$, and $\gamma/2\pi$ are the internal uniaxial anisotropy field, the saturation magnetization, and the gyromagnetic ratio of the SMF, respectively. Many studies focused on the enhancement of $H_K$ because it is not only easier to control by various means, but also has a larger amplitude of change than $4\pi M_S$. Many useful methods, such as using oblique sputtering [12–14], doping element composition gradient sputtering (CGS) [7,15,16], post-annealing in magnetic field [17], and magnetoelectric coupling effect [18–20], have been proposed. After several decades of efforts, the self-biased $f_r$ of SMFs was improved from radiofrequencies up to microwave frequencies [9,21–26].

In recent years, optical mode (OM) resonance based on interlayer exchange coupling (IEC) has received substantial attention [27,28]. The very high effective field ($H_{IEC}$) of IEC shows good potential to drive the OM FMR up to several tens of GHz [29–36]. For exchange-coupled sandwich films, the total free energy includes an exchange energy term [37].

$$\varepsilon_{ex} = -J_1 \frac{\vec{M}_A \cdot \vec{M}_B}{M_A M_B} - J_2 \left( \frac{\vec{M}_A \cdot \vec{M}_B}{M_A M_B} \right)^2 , \qquad (2)$$

where $\vec{M}_A$ ($\vec{M}_B$) are the magnetization vectors of layer $A$ ($B$). $J_1$ and $J_2$ refer to the bilinear and biquadratic coupling coefficient, respectively. If $J_1 > 0$, ferromagnetic (FM) coupling occurs, whereas, if $J_1 < 0$ (when $J_2 < 0$), antiferromagnetic (AFM) coupling dominates [38]. In general, two resonance modes, the acoustic and optical modes, appear in FM/NM/FM systems [39]. The coupling strength and type are sensitive to the material type and film thickness of the FM and NM layers [40]. Parkin et al. extensively studied the effects of nonmagnetic spacer elements on strength and oscillation of the RKKY coupling [41]. Very recently, an ultrahigh OM resonance frequency over 22 GHz was obtained in exchange-coupled FeCoB/Ru/FeCoB sandwich structures with magnetic uniaxial anisotropy in our laboratory [42]. In order to further explore the effect of non-magnetic spacers (such as Ta, Hf, W, Cr, Si, etc.) on the IEC, in this study, the effects of the hafnium thickness on the IEC and high-frequency FMR performances are reported.

## 2. Experimental

A Si (100) substrate with a size of $80 \times 5 \times 0.5$ mm$^3$ was pasted on the sample holder with the sample's length direction along the arrangement direction of boron pieces (see Figure 1 inset, the R direction). The substrate was cut into 5 mm $\times$ 5 mm pieces along the R direction for magnetic and microwave measurements, sequentially labeled as sample positions 1–16 (Figure 1). The FeCoB (25 nm)/Hf ($t$ nm)/FeCoB (25 nm) trilayers (named as TL) were prepared by RF magnetron sputtering on Si substrates under a vacuum less than $5 \times 10^{-5}$ Pa. Control samples of a 50 nm FeCoB single layer without Hf spacer were also deposited (labeled SL). The FeCoB sublayers of the trilayer were deposited by a modified compositional gradient sputtering (MCGS) method using a boron-pasted FeCo target (as shown in Figure 1, several boron pieces with dimensions of $5 \times 5 \times 0.5$ mm$^3$ were put on the surface of the $Fe_{0.7}Co_{0.3}$ target) under an RF power of 60 W, deposition argon pressure of 0.5 Pa at an Ar flow rate of 60 sccm, and a holder rotating speed of 10 rpm [16]. In the conventional CGS method, the substrate is kept stationary without rotation. A boron gradient distribution along the R direction is formed, which results in uniaxial compressive stress along the R direction. As a result, a stress-induced uniaxial magnetic anisotropy field with an easy axis along the T direction is obtained for the FeCoB film with a positive magnetostriction coefficient [42]. Because the boron concentration and gradient are different for the different test positions, the magnetic properties of the CGS sample show an evident position dependence. In this study, the FeCo target center was set to point to the center of the substrate as in the conventional CGS method, but the substrate was rotated during the deposition. Using this modified CGS (MCGS) method, magnetic anisotropic films are also available, but their position dependence is weakened. As a result, a position-insensitive anisotropy is formed in the FeCoB films with the easy (hard) axis perpendicular (parallel) to the R direction, respectively, similar to previous studies [15,16]. In this study, some almost sample-position-independent magnetic resonance spectra were obtained for the SL along the easy axis (EA). Next, a wedge Hafnium spacer with thickness increasing along R was deposited by oblique sputtering on the bottom 25 nm FeCoB sublayer at 45 W $\times$ 75 s under an Ar pressure of 0.6 Pa. In the oblique sputtering method, the hafnium target maintains a certain oblique angle with respect to the Si substrate plane, i.e., the hafnium target center was set to point to the position #16 end of the substrate, and the turntable was kept stationary (no rotation) during the deposition of hafnium. As a result, we obtained a wedge hafnium spacer with an almost linear increment of hafnium thickness from position #1 to #16 along the R direction (Figure 2). The hafnium thickness distribution along the R direction was calibrated with a thicker hafnium film by atomic force microscopy (AFM, Parkin EX7). Thus, the thickness of the actual Hafnium spacer at each position can be calculated from the calibrated deposition rate and the actual deposition time. The static

magnetic properties of the films were carried out by an alternating gradient magnetometer (AGM, MicroMagTM 2900, Princeton Measurements Corporation, Princeton, NJ, USA).

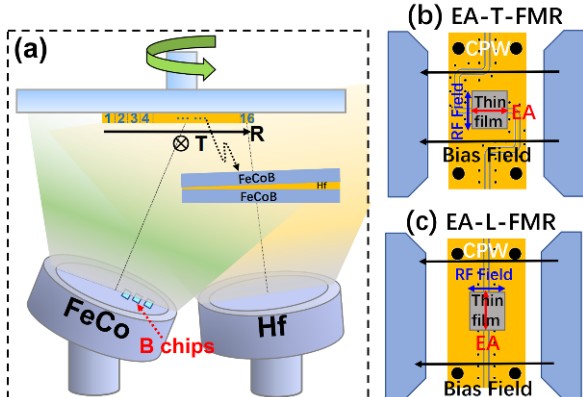

**Figure 1.** (**a**) Schematics of the MCGS device and the profiles of the film, fixture, and magnetic field for the EA-T-FMR (**b**) and the EA-L-FMR (**c**). The inset in (**a**) shows the structure of the FeCoB/Hf/FeCoB trilayer.

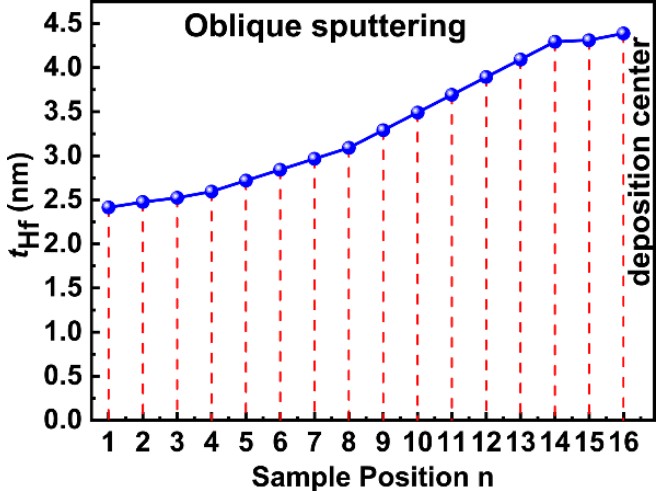

**Figure 2.** Sample position *n* dependence of the calculated Hafnium thickness $t_{Hf}$.

The dynamic high-frequency performance of the films was characterized by a vector network analyzer (VNA, N5224A, Agilent Technologies Co., Ltd., Santa Clara, CA, USA) with a coplanar waveguide fixture. Both the transverse and the longitudinal FMR measurement modes were adopted to distinguish the mode type [43]. The transverse pumping FMR (T-FMR) is sensitive to the AM resonance (Figure 1b), while the longitudinal pumping FMR (L-FMR) is very sensitive to the OM resonance (Figure 1c) [39,43–45]. The resonance mode can be easily identified from the relative intensities of the resonance peaks. The EA-T-FMR refers to the transverse FMR mode with wave vector (transmission line) along the EA direction, while EA-L-FMR refers to the longitudinal FMR mode with wave vector along the EA direction (Figure 1b,c). The FMR response was carried out by frequency scanning in the 1–15 GHz range at a certain field *H*, before increasing *H* and repeating the measurement process. The variation range of the field was 0–1000 Oe with an increasing step of 5 Oe. The two-dimensional contour projection diagram (2D color $S_{21}$–*H* diagram) could be carried out for T- or L-mode measurements along the easy or hard axis of the sample, revealing the relationship between acoustic mode (AM) and OM resonance with the applied field, which also helped us to judge the assignment of AM or OM resonance to each $S_{21}$–*f* curve. In addition, it is possible that the coplanar waveguide fixture may generate a strong magnetic field at higher microwave power, thus interfering with the measurement results. To avoid this, a very weak microwave power of −30 dB was adopted,

which generated a magnetic field less than several mOe around the transmission line of the fixture.

The polar diagrams of $S_{21}$ ($S_{21}-\theta$ curve) and frequency ($f_r-\theta$ curve) were measured by rotating the film around its normal axis. The value of $S_{21}$ is proportional to the total magnetization in the measurement direction. Thus, the $S_{21}-\theta$ curve can reveal the distribution of magnetic moments in the film. In this study, $\theta$ refers to the angle between the measurement direction and the easy axis, i.e., $\theta = 0°$ along the EA.

## 3. Results and Discussion

As illustrated in Figure 2, the hafnium thickness increased monotonically as the test position approached the deposition center of the hafnium target. This indicates that hafnium thickness can be used as a variable to study its influence on IEC. The hafnium thicknesses of the last few points varied slowly, because they were close to the central region of the hafnium sputtering, where the deposition was relatively uniform. Figure 3 compares the magnetic spectra between FeCoB SL and FeCoB/Hf/FeCoB TL, where $S_{21}$ refers to the scattering parameter of microwaves from port 1 to port 2 through the sample. As expected, the spectra of all SL segments were almost identical (Figure 3a). In contrast, the TL magnetic spectra showed a strong position (hafnium thickness) dependence (Figure 3b). For the segments with $n < 8$ or $n > 14$, only one resonance peak was observed at zero external field along EA, while two peaks were present for the segments with $n$ between 8 and 13. These facts imply that certain hafnium thickness-dependent interlayer coupling occurs. Comparing the FMR frequency between SL and TL, the TL showed improved frequencies. For example, the upper FMR frequency of the $n = 9$ trilayer was 4.87 GHz, while that of the SL was around 3.3 GHz. An enhancement of 1.57 GHz with an increment ratio of 48% was achieved.

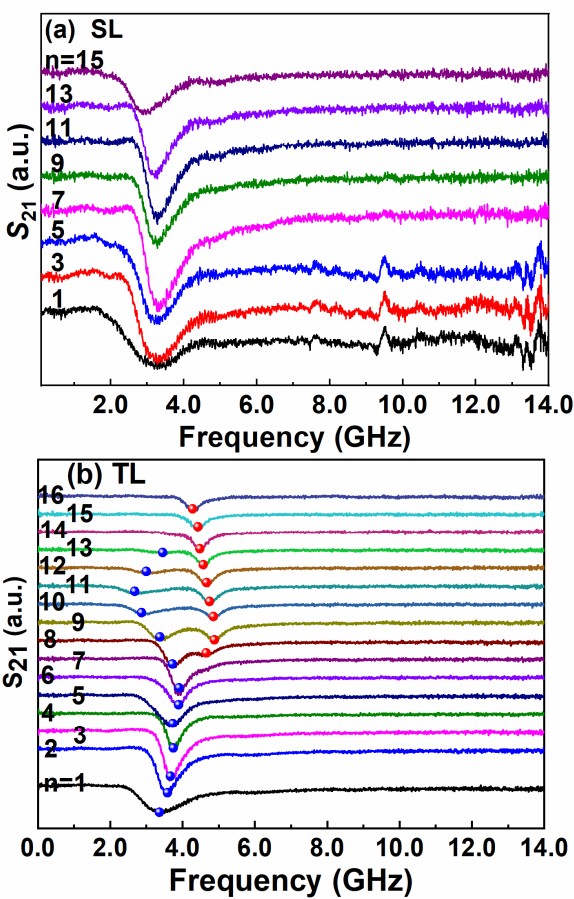

**Figure 3.** The $S_{21}-f$ curves along EA at zero external field at various test positions $n$ for (**a**) single layer and (**b**) trilayer.

The angle-dependent polar diagrams of FMR frequency and $S_{21}$ amplitude at zero magnetic field, shown in Figure 4, revealed three types of in-plane anisotropy distributions. For the thinner hafnium spacer sample ($n = 7$), only one set of FMR peaks was observed with a frequency distribution similar to a pair of parentheses (Figure 4a), and an "8"-shaped $S_{21}$–$\theta$ curve was obtained (Figure 4b). These facts indicate an obvious uniaxial magnetic anisotropy with the easy axis along $0°$ in the film (transverse to the R direction in the deposition). For the sample with medium hafnium thickness ($n = 10$), two set of FMR peaks appeared, forming a double-8 shape (Figure 4c,d), indicating that two resonance modes were present, but bilinear interlayer coupling was still dominant with the easy axis along $0°$. In contrast, for the thicker hafnium spacer sample ($n = 16$), a nearly circular frequency polar diagram ($f_r$–$\theta$ curve) was observed (Figure 4e), and the $S_{21}$ intensity (or magnetic moments) was distributed along two orthogonal axes, suggesting an important contribution of biquadratic coupling (Figure 4f).

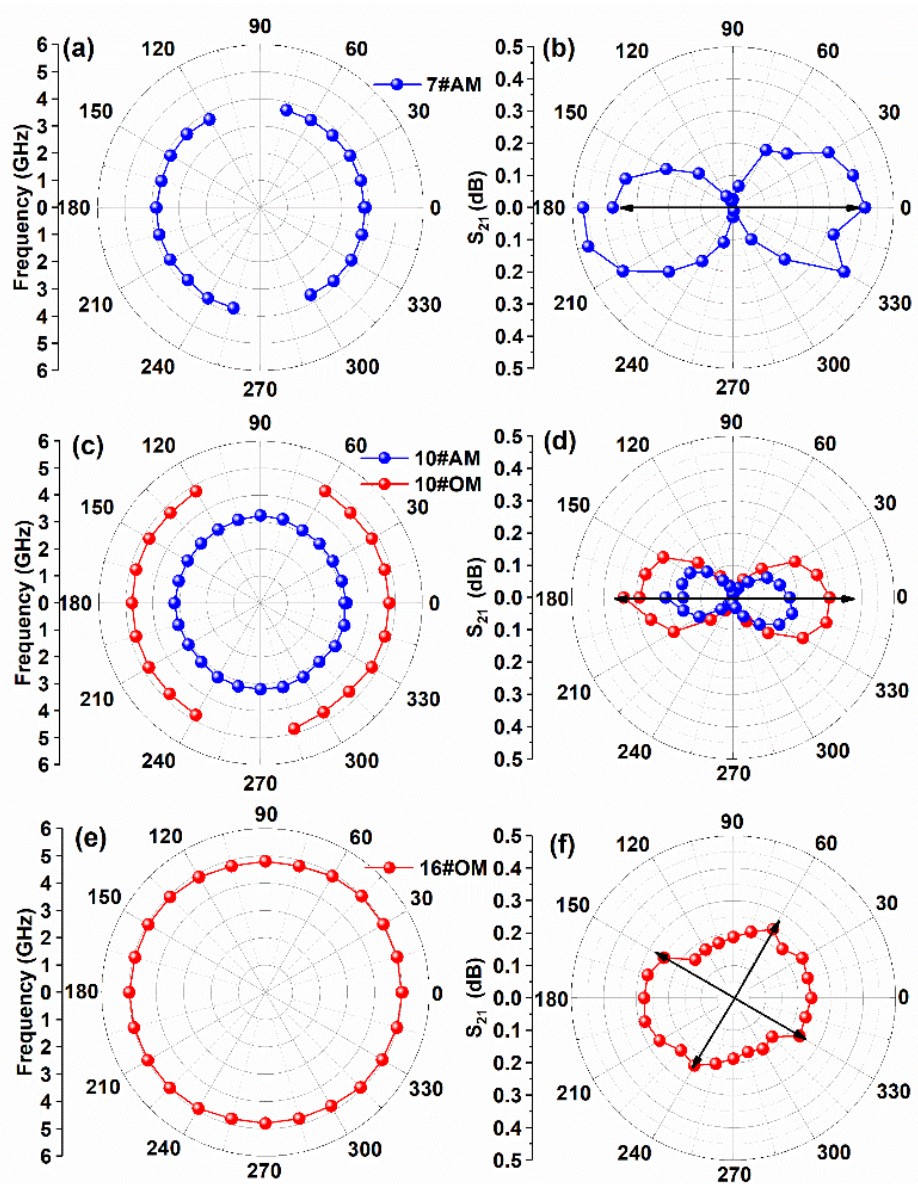

**Figure 4.** The polar diagrams of FMR frequency (left panels) and $S_{21}$ amplitude (right panels) at zero external field for three representative samples with $n = 7$ (**a,b**), 10 (**c,d**), and 16 (**e,f**), respectively.

Figure 5 shows the $S_{21}$–$H$ curves for typical samples along the EA direction. From Figure 5a,b, it can be seen that, for the sample $n = 6$ with a thinner hafnium spacer, two $f$–$H$ lines were parallel and nearly overlapped in the EA-T-FMR mode, and only

one very weak AM trace vanishing at 40 Oe was observed in the EA-L-FMR mode. For the sample $n = 8$ with a medium hafnium thickness, two separated $f$–$H$ lines existed in EA-T-FMR, while AM and OM resonance modes were both present in EA-L-FMR up to 112 Oe for OM. In contrast, for the sample $n = 16$ with a thicker hafnium spacer, two obviously separated $f$–$H$ lines were observed in the EA-T-FMR mode, with the lower branch showing a "$\sqrt{\phantom{x}}$"-shape, and only an OM line was present in the EA-L-FMR mode up to 178 Oe. The lower resonance frequency of AM than OM demonstrates that they were AFM coupled [46–48].

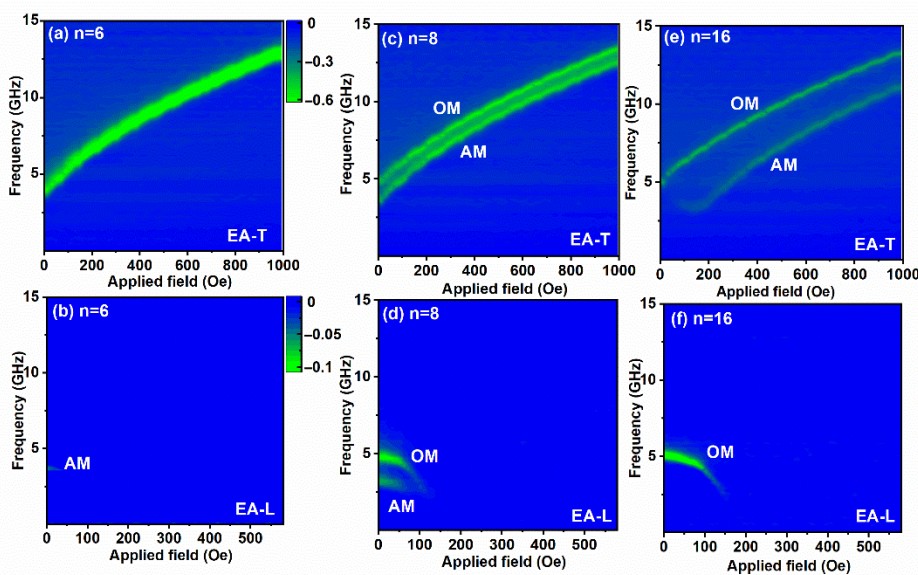

**Figure 5.** The 2D contour projection diagrams ($S_{21}$–$H$ curves) for typical samples with $n = 6$ (**a**,**b**), 8 (**c**,**d**), and 16 (**e**,**f**) measured at T- and L-modes along EA direction.

In order to clarify the mechanism of interlayer coupling, static and dynamic magnetic performance was studied. Figure 6 shows the hysteresis loops of representative samples. As illustrated, the hysteresis loops of the samples with $n \leq 8$ showed an obvious EA and HA, and the remanence ratio $m_r$ of the EA loop was close to 1 (Figure 6a,b). The loops looked like one ferromagnetic phase, suggesting that bilinear interlayer coupling existed in the sample. However, for the samples with $n \geq 10$, the $m_r$ of the EA loops decreased, while that of the HA loops increased. Furthermore, $m_r$ of EA and HA in the $n = 16$ sample had almost the same values, and the HA loop was more easily saturated than the EA loop (Figure 6c–e). This fact indicates that its coupling type was evidently different from the thinner hafnium samples, and a strong biquadratic coupling was present in this sample. The anisotropy fields $H_K$ were extracted from the hysteresis loops, as summarized in Figure 6f, and they clearly demonstrate the variation of interlayer coupling strength with the hafnium thickness.

Layadi's rigid model was adopted to further understand the theoretical origin of both resonance modes [49]. This model assumes that the magnetizations of the two ferromagnetic layers can be represented as single magnetization vectors $M_A$ and $M_B$, which are lying in the film plane. Layadi's model is not applicable to general polycrystalline trilayers due to the multidomain structure with random orientation of magnetization. However, in this study, the FeCoB/Hf/FeCoB trilayer exhibited an IEC and a uniaxial magnetic anisotropy, and the magnetizations in magnetic layers were along the EA direction. Therefore, the magnetization of the whole trilayer can be considered as an orientating along the same axis, conforming to the requirement of Layadi's model. It was assumed that the $x$–$y$ plane was the film plane, while the normal direction of the film plane was along the $z$-axis (see Figure 1b). The EA and applied field $H$ directions were set are along the $x$-axis ($\phi_H = 0°$), and the microwave field $\widetilde{h}$ was set along the $y$-axis. The total free energy included the Zeeman energy, the in-plane uniaxial anisotropy energy (with constant $K_i$, $i = A$ or $B$),

the shape anisotropy, any out-of-plane uniaxial magnetocrystalline anisotropy energy ($K_{ueff} = K_u - 2\pi M^2$, where $K_u$ is the anisotropy constant), and interlayer coupling energy. The total free energy per unit area can be described as follows [50]:

$$
\begin{aligned}
E = {}& t_A \left\{ -M_A H \sin\theta_A \cos\phi_A + K_{ueffA} \sin^2\theta_A - K_A \sin^2\theta_A \cos^2\phi_A \right\} \\
&+ t_B \left\{ -M_B H \sin\theta_B \cos\phi_B + K_{ueffB} \sin^2\theta_B - K_B \sin^2\theta_B \cos^2\phi_B \right\} \\
&- J_1 \left\{ \sin\theta_A \sin\theta_B \cos(\phi_A - \phi_B) + \cos\theta_A \cos\theta_B \right\} \\
&- J_2 \left\{ \sin\theta_A \sin\theta_B \cos(\phi_A - \phi_B) + \cos\theta_A \cos\theta_B \right\}^2
\end{aligned}
\tag{3}
$$

where $t_A$ and $t_B$ refer to the thicknesses of ferromagnetic layers $A$ and $B$, respectively. $\theta_A$ ($\theta_B$) and $\phi_A$ ($\phi_B$) are the polar and azimuthal angles of $M_A$ ($M_B$), respectively. The resonance frequency can be obtained as follows [49,50]:

$$
\begin{aligned}
\left[\frac{ab}{\gamma_A\gamma_B}\right]^2 \omega^4 &- \left[ a^2 b^2 \left( \frac{H_1^A H_2^A}{\gamma_B^2} + \frac{H_1^B H_2^B}{\gamma_A^2} \right) + abc_1 \left( \frac{aH_2^B}{\gamma_A^2} + \frac{bH_2^A}{\gamma_B^2} \right) \right. \\
&\left. + abc_2 \left( \frac{aH_1^B}{\gamma_A^2} + \frac{bH_1^A}{\gamma_B^2} \right) + c_1 c_2 \left( \frac{a^2}{\gamma_A^2} + \frac{b^2}{\gamma_B^2} \right) + \frac{2c_0 c_2 ab}{\gamma_A\gamma_B} \right] \omega^2 \\
&+ \left[ abH_2^A H_2^B + c_2 \left( aH_2^A + bH_2^B \right) \right] \times \left[ abH_1^A H_1^B + \right. \\
&\left. c_1 \left( aH_1^A + bH_1^B \right) + \left( c_1^2 - c_0^2 \right) \right] = 0,
\end{aligned}
\tag{4}
$$

where $a = t_A M_A$, $b = t_B M_B$, and $\gamma_A/2\pi = \gamma_B/2\pi = 2.8$ GHz/kOe. According to the rigid model, the angle $\phi_A$ of the rigid layer is fixed at a certain value, and the parameters $c_j$ and $H_j^i$ are expressed as follows:

$$
c_0 = J_1 + 2J_2 \cos(\phi_A - \phi_B)
$$

$$
c_1 = J_1 \cos(\phi_A - \phi_B) + 2J_2 \cos^2(\phi_A - \phi_B)
$$

$$
c_2 = J_1 \cos(\phi_A - \phi_B) + 2J_2 \cos 2(\phi_A - \phi_B)
$$

$$
H_1^A = H \cos\phi_A - H_{KeffA} + H_{KA} \cos^2\phi_A
$$

$$
H_2^A = H \cos\phi_A + H_{KA} \cos 2\phi_A
$$

$$
H_1^B = H \cos\phi_B - H_{KeffB} + H_{KB} \cos^2\phi_B,
$$

$$
H_2^B = H \cos\phi_B + H_{KB} \cos 2\phi_B
$$

where $H_{KeffA} = 2K_{ueffA}/M_A$, $H_{KeffB} = 2K_{ueffB}/M_B$, $H_{KA} = 2K_A/M_A$, and $H_{KB} = 2K_B/M_B$ are the planar anisotropy and the effective uniaxial fields for layers A and B, respectively.

Since the AM and OM resonances merged together for the samples at $n < 6$ due to the strong ferromagnetic interlayer coupling, we focused on the interlayer coupling for samples at $n = 7$–16. Figure 7a shows the comparison of $f_r$–$H$ curves between the samples at $n = 7$–16. As illustrated, the FMR frequencies of OM were very close to each other (Figure 7a) with a slight increase-then-decrease trend with hafnium thickness (Figure 7a inset). The AM FMR frequencies at different positions separated from each other, shifting to a lower frequency with the increase in hafnium thickness. Figure 7b,c show the experimental and fitted data using Equation (3) for two representative trilayers. Both $M_A$ and $M_B$ were deduced from the hysteresis loops, but the in-plane uniaxial anisotropy fields $H_K$ were affected by the thickness of hafnium spacers (Figure 6f). For the thinner hafnium sample ($n = 7$), only bilinear coupling occurred, and the biquadratic coupling coefficient $J_2$ could be neglected. As shown in Figure 7b, the simulated and experimental results were consistent with each other, and an AFM coupling with $J_1 = -1.27$ merg/cm$^2$ was obtained. However, for the thicker hafnium sample ($n = 16$), the biquadratic coupling increased; thus, $J_1$ and $J_2$ were considered together. As shown in Figure 7c, $J_1$ and $J_2$ of $-0.83$ and $-0.15$ merg/cm$^3$

were obtained, respectively. The $J_1$, $J_2$, and effective coupling coefficient $J_{eff}$ ($=J_1 + 2J_2$) for trilayers at $n = 7$–16 are summarized in Figure 7d. As illustrated, the interlayer coupling was relatively weak, and $J_1$ varied in a small range of $-0.8$ to $-1.7$ merg/m². For the samples with $n < 10$, bilinear coupling dominated, and $J_1$ slightly increased. With the increase in hafnium thickness, biquadratic coupling appeared and increased, while $J_1$ decreased. The decrease in AM resonance frequency with hafnium thickness shown in Figure 7a could be attributed to the contribution of $J_2$ [49,50].

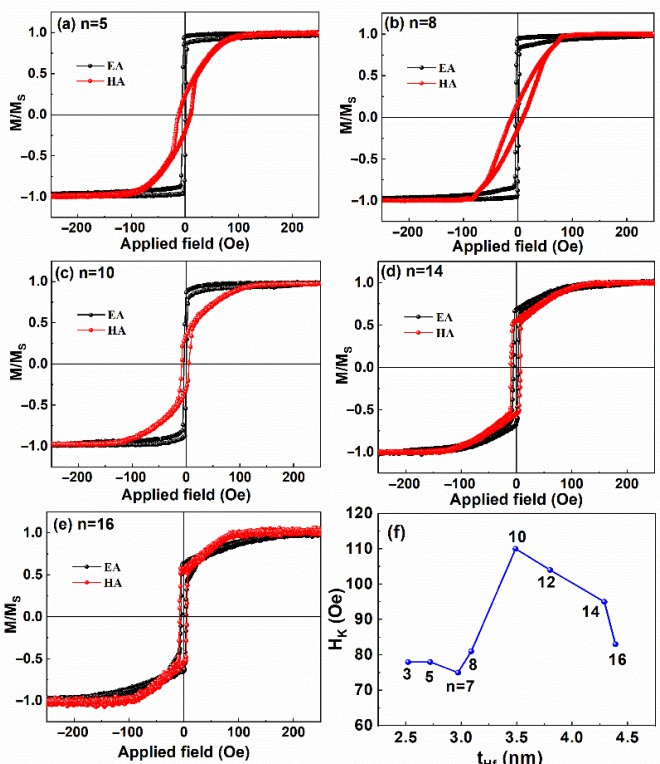

**Figure 6.** (**a**–**e**) The EA and HA hysteresis loops for trilayer segments at various positions; (**f**) the variation of anisotropy field $H_K$ with the hafnium thickness.

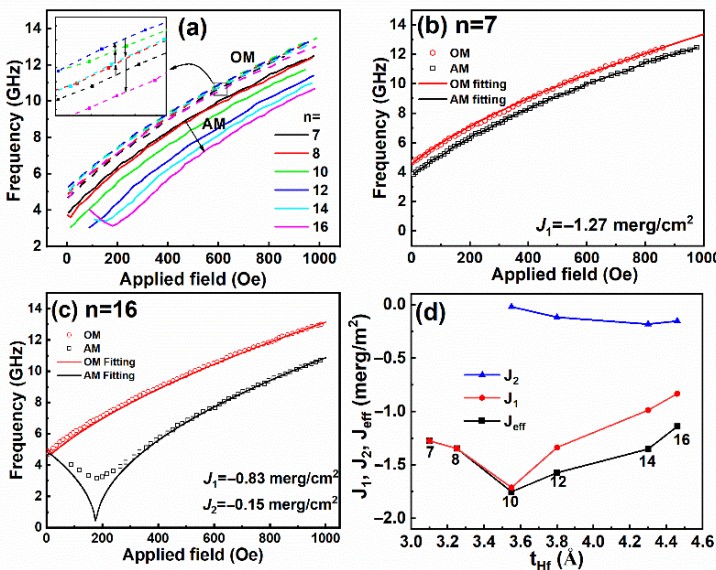

**Figure 7.** (**a**) The *f*–*H* curves of samples at $n = 7$–16 in EA-T mode; (**b**,**c**) the experimental and simulation results for representative samples at $n = 7$ and 16, respectively; (**d**) the hafnium thickness dependence of interlayer coupling coefficients for samples at $n = 7$–16.

For the thinner hafnium spacer with $t_{Hf}$ < 3 nm, only FM coupling was observed, potentially due to the interface roughness, pinhole effect, etc., while, for the trilayers with $t_{Hf}$ > 3 nm, AFM coupling appeared. In general, the strength of the Ruderman–Kittel–Kasuya–Yosida (RKKY) interaction decays beyond a typical thickness of 1 nm. It is unexpected that interlayer coupling occurs in a trilayer with a nonmagnetic spacer thicker than 3 nm. However, in 2013, Li et al. found that the interlayer exchange coupling of the Co (5 Å)/Pt($t_{Pt}$ Å)/Co (5 Å) multilayers oscillated between AFM and FM as a function of Pt spacer thickness for a Pt spacer thickness of $t_{Pt}$ > 3.4 nm. The good fitting of the experimental data and simulated curves showed that the RKKY-type coupling was dominant in the multilayers [51]. A similar phenomenon was also observed in trilayers with thicker nonmagnetic spacers, such as FeCoB/Ta/FeCoB with $t_{Ta}$ > 3 nm and FeCoB/Ru/FeCoB with $t_{Ru}$ > 2.5 nm (not shown here). These studies indicate that RKKY-type coupling in a sandwich trilayer may be present in a relatively thicker nonmagnetic spacer. This interesting phenomenon is expected to be further studied theoretically in the future. On the other hand, good high-frequency performance appeared at a hafnium thickness of 3–4 nm, which is relatively thicker and easier to fabricate. Compared with the ruthenium spacer, its good performance exists in the extremely low thickness of ca. 0.3 nm [29], for which sample preparation is very strict and difficult. Thus, the Hf separated trilayers are beneficial for practical application.

## 4. Conclusions

The effect of hafnium spacer thickness on the interlayer coupling and ferromagnetic resonance was systematically investigated in modified compositional gradient-sputtered FeCoB (25 nm)/Hf($t_{Hf}$)/FeCoB (25 nm) trilayers. It was revealed that the interlayer coupling and FMR resonance were sensitive to the hafnium thickness, while the FMR frequency was enhanced by 48% due to interlayer coupling. Ferromagnetic coupling between the FeCoB films occurred for the trilayers with $t_{Hf}$ < 3.0 nm, which could be attributed to the interface roughness, pinhole effect, etc. A hafnium thickness-dependent interlayer coupling appeared in the trilayers at $t_{Hf}$ > 3.0 nm, while biquadratic coupling appeared at $t_{Hf}$ > 3.5 nm. This study demonstrates that AFM coupling may exist in the FM/NM/FM trilayer with a thicker nonmagnetic spacer, leading to the desired higher-frequency OM FMR.

**Author Contributions:** D.L., investigation, writing—original draft, and data curation; S.Z., investigation and data curation; S.L., supervision, conceptualization, methodology, and project administration. All authors have read and agreed to the published version of the manuscript.

**Funding:** This work was financially supported by the National Natural Science Foundation of China with grant No. 51871127 and 11674187.

**Institutional Review Board Statement:** Not applicable.

**Informed Consent Statement:** Not applicable.

**Data Availability Statement:** Data available on request from the corresponding author.

**Conflicts of Interest:** The authors declare no conflict of interest.

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
