# Peer review of "Interlayer Coupling and High-Frequency Performance in Magnetic Anisotropic FeCoB/Hf/FeCoB Trilayers with Various Hf Thicknesses"

_magnetochemistry, doi:10.3390/magnetochemistry8060065_

Round 1

Reviewer 1 Report

This paper presents some experimental data about ferromagnetic resonant frequency of FeCoB/Hf/FeCoB trilayers and consideration to explain the phenomena using Layadi’s rigid model. I have following questions and comments indicated below.

[questions and comments]

1.

Although the FeCoB films were prepared on rotating substrate stage, the FeCoB films has a magnetic anisotropy field which is expressed as Hk lying in the T direction in Fig.1(a). Since the anisotropy field is important to discuss FMR study, the origin and reproducibility of the anisotropy should be clarified and/or noted. Since schematic configuration of deposition method seems to be different from that indicated in Ref.[15-16], it is difficult to understand the origin of anisotropy in FeCoB films.

2.

Layadi’s rigid model used in this study assume that the magnetization of the two ferromagnetic layers can be represented as single magnetization vectors MA and MB. and they are lying in the film plane. However there is a possibility that the magnetization configuration in the two magnetic layers are not homogeneous, such as taking the magnetic domain configuration and/or appearance of perpendicular magnetization component. Especially for the trilayer with thick Hf layer shows the M-H loops which is not appropriate to be regarded as single magnetization domains.

Authors should make comments of availability to adapt Layadi’s rigid model to explain the results.

3. [L.184-186, L.264-271]

As authors have already commented that it is difficult to explain the interaction between two ferromagnetic layers, such as magnetostatic and RKKY interactions, enhances as the interlayer(Hf) thickness increases. I think it requires another explanation to account the experimental results.

[Typographical corrections (suggestion)]

L.215

θ are disappeared in the sentence.

Please add comment for θA and θB, they are regarded as 90 deg. for the calculation performed in this study.

L.148

“… segments with n between 9 and 14.”

It seems to be better “… segments with n between 8 and 13.”

Author Response

Response to the Reviewer 1#

This paper presents some experimental data about ferromagnetic resonant frequency of FeCoB/Hf/FeCoB trilayers and consideration to explain the phenomena using Layadi’s rigid model. I have following questions and comments indicated below.

[questions and comments]

Q1. Although the FeCoB films were prepared on rotating substrate stage, the FeCoB films has a magnetic anisotropy field which is expressed as Hk lying in the T direction in Fig.1(a). Since the anisotropy field is important to discuss FMR study, the origin and reproducibility of the anisotropy should be clarified and/or noted. Since schematic configuration of deposition method seems to be different from that indicated in Ref. [15-16], it is difficult to understand the origin of anisotropy in FeCoB films.

Response: In our laboratory, there are two kinds of CGS methods. For the common CGS method, the substrate is kept stationary without rotation. A Boron gradient distribution along R direction is formed, which results in a uniaxial compressive stress along R direction. As a result, a stress-induced uniaxial magnetic anisotropy field with easy axis along T direction is obtained for the FeCoB film with a positive magnetostriction coefficient. Because the Boron concentration and gradient are different for the different test positions, the magnetic properties of the CGS sample show an evident position-dependence. However, in this study, a modified CGS method was adopted, where the FeCo target center was set to point to the center of the substrate as usual CGS method, but the substrate was rotating, instead of stationary, during the deposition. With this MCGS method, magnetic anisotropic films are also available but their position dependence is weakened. As a result, a position insensitive anisotropy is formed in the FeCoB films with the easy (hard) axis perpendicular (parallel) to the R direction, respectively.

The description and comparison between common CGS method and MCGS method were added in revision lines 60-65.

Q2. Layadi’s rigid model used in this study assume that the magnetization of the two ferromagnetic layers can be represented as single magnetization vectors MA and MB. and they are lying in the film plane. However, there is a possibility that the magnetization configuration in the two magnetic layers are not homogeneous, such as taking the magnetic domain configuration and/or appearance of perpendicular magnetization component. Especially for the trilayer with thick Hf layer shows the M-H loops which is not appropriate to be regarded as single magnetization domains. Authors should make comments of availability to adapt Layadi’s rigid model to explain the results.

Response: Layadi's rigid model is an ideal model, where the magnetization of the two ferromagnetic layers can be represented as single magnetization vectors MA and MB. and they are lying in the film plane. Layadi’s model is not applicable to general polycrystalline trilayers due to the multi-domain structure with random orientation of magnetization. However, in this study, the FeCoB/Hf/FeCoB trilayer exhibits an interlayer exchange coupling and a uniaxial magnetic anisotropy, the magnetizations in upper or lower magnetic layers are along the easy axis direction. Therefore, the magnetization of the whole trilayer can be considered as orientating along the same axis, which is conformed to the requirement of Layadi’s model. Of course, there may be a possibility that the magnetization configuration in the two magnetic layers are not homogeneous, such as taking the magnetic domain configuration and/or appearance of perpendicular magnetization component. But we believed that for the samples in this study, the inhomogeneity of two magnetic layers can be omitted. The model can describe the actual situation basically.

The availability of Layadi’s rigid model was appended in the revision lines 161-167.

Q3. [L.184-186, L.264-271] As authors have already commented that it is difficult to explain the interaction between two ferromagnetic layers, such as magnetostatic and RKKY interactions, enhances as the interlayer (Hf) thickness increases. I think it requires another explanation to account the experimental results.

Response: The present of interlayer exchange coupling in a trilayer with a thicker non-magnetic spacer is unexpected. Nevertheless, we have recently found that the same phenomenon in FeCoB/Ta/FeCoB trilayer with Ta thickness thicker than 3 nm. Moreover, similar phenomenon also occurs when Ru is thicker than 2.5 nm in FeCoB/Ru/FeCoB trilayers. In 2013, Li L. et al. found that RKKY-type coupling still exists when Pt thickness is greater than 3.4 nm. Therefore, it is believed that there exists interlayer coupling between ferromagnetic layers through relatively thicker non-magnetic spacer. These interesting studies may change our conventional understanding and require further theoretical and experimental research.

[Typographical corrections (suggestion)]

Q4. L.215 θ are disappeared in the sentence. Please add comment for θA and θB, they are regarded as 90 deg. for the calculation performed in this study.

Response: Thank you for your remind. It has been corrected.

Q5. L.148 “… segments with n between 9 and 14.” It seems to be better “… segments with n between 8 and 13.”

Response: I agree with you. They are modified.

Reviewer 2 Report

Results were clearly presented to support the findings of the study. 

Author Response

No comments.

Reviewer 3 Report

The manuscript deals with experimental investigation of ferromagnetic and antiferromagnetic resonanses in RKKY-coupled magnetic trilayers. Studying magnetic resonances in exchange coupled multiilayers has been reported for many years, since magnetic resonance is applicable to determinig the layer-coupling parameters. Using a specific interlayer spacer (Hf) between technologically important magnets (FeCoB) seems to be novel. I find the manuscript to be logically written and of potential interest. However, it requires improvements in places.

1. The description of the deposition method is difficult to understand. Adding to fig. 1 a scheme of the deposited structure could help.

2. The scattering parameters S_21 should be named in captions to figs. 3-5. 

3. The hysteresis curves in fig. 6 should be compared to theoretical ones, obtainable within the Stoner approach. The problem is old and I guess the relevant literature is available (please check: Guo Guang-Hua et al 2008 Chinese Phys. Lett. 25 2634 for instance, etc.).

4. In Conclusins section, please provide a comparison of the advantage (efficiency) of the coupling through the Hf interlayer to the previously investigated couplings through other interlayers (Ta, Ru,...). The last point seems to be crucial for motivating the study.

Author Response

Response to the Reviewer 3#

 The manuscript deals with experimental investigation of ferromagnetic and antiferromagnetic resonanses in RKKY-coupled magnetic trilayers. Studying magnetic resonances in exchange coupled multiilayers has been reported for many years, since magnetic resonance is applicable to determing the layer-coupling parameters. Using a specific interlayer spacer (Hf) between technologically important magnets (FeCoB) seems to be novel. I find the manuscript to be logically written and of potential interest. However, it requires improvements in places.

Q1. The description of the deposition method is difficult to understand. Adding to fig. 1 a scheme of the deposited structure could help.

Response: A scheme of the structure of trilayer has been added in the Figure 1a inset.

Q2. The scattering parameters S21 should be named in captions to figs. 3-5.

Response: The definition of S21 was added. 

Q3. The hysteresis curves in fig. 6 should be compared to theoretical ones, obtainable within the Stoner approach. The problem is old and I guess the relevant literature is available (please check: Guo Guang-Hua et al 2008 Chinese Phys. Lett. 25 2634 for instance, etc.).

Response: Thank you very much for your valuable suggestion. The theoretical analysis of hysteresis loop is very important, but we are not familiar with the theoretical calculation. I am afraid that we cannot finish the calculation in the short revision time of 5 days. I will study hard and master the calculation method of hysteresis loop as soon as possible.

Q4. In Conclusions section, please provide a comparison of the advantage (efficiency) of the coupling through the Hf interlayer to the previously investigated couplings through other interlayers (Ta, Ru,...). The last point seems to be crucial for motivating the study.

Response: The good high-frequency performance appears at the Hf thickness of 3-4 nm, which is relatively thicker, and easier to fabricate. Comparing with the Ru spacer, its good performance exists in the extremely thin thickness of c.a. 0.3 nm, so the sample preparation is very strict and difficult. So the Hf separated trilayers are beneficial for practical application. The corresponding discussion was added in lines 221-223.
